# Network QoS Impact on Spatial Perception through Sensory Substitution in Navigation Systems for Blind and Visually Impaired People

**DOI:** 10.3390/s23063219

**Published:** 2023-03-17

**Authors:** Santiago Real, Alvaro Araujo

**Affiliations:** B105 Electronic Systems Lab, Escuela Técnica Superior de Ingenieros de Telecomunicación, Universidad Politécnica de Madrid, Avenida Complutense 30, 28040 Madrid, Spain

**Keywords:** QoS, sensory substitution, spatial perception, spatial cognition, navigation, virtual reality, orientation and mobility, wearable

## Abstract

A navigation system for individuals suffering from blindness or visual impairment provides information useful to reach a destination. Although there are different approaches, traditional designs are evolving into distributed systems with low-cost, front-end devices. These devices act as a medium between the user and the environment, encoding the information gathered on the surroundings according to theories on human perceptual and cognitive processes. Ultimately, they are rooted in sensorimotor coupling. The present work searches for temporal constraints due to such human–machine interfaces, which in turn constitute a key design factor for networked solutions. To that end, three tests were conveyed to a group of 25 participants under different delay conditions between motor actions and triggered stimuli. The results show a trade-off between spatial information acquisition and delay degradation, and a learning curve even under impaired sensorimotor coupling.

## 1. Introduction

The penetration of pervasive computing in society has opened new possibilities as to how we interact with the world. For instance, smartphones and low-cost wearables serve as a window to an ever-increasing amount of information, backed by a network that manages the acquisition, storage, and processing of data.

This new paradigm is promoting the development of devices that assist individuals with perceptual and cognitive deficiencies in everyday tasks [1]. These devices gather real-time data of the environment and provide the user with key information to alleviate disabilities, e.g., assisting social interactions in cases of Alzheimer’s by providing information tips or guiding visually impaired individuals throughout a route. Ongoing research on this topic focuses on collecting user needs under specific scenarios and translating these to device design.

Although the specific technical requirements vary according to the deficiency and the type of assistance provided, system delay was revealed as a key factor in tasks that required human–environment interactions. Additionally, commercially available devices usually need to be portable and lightweight while withstanding hours of uninterrupted operation under relatively high computational load. Therefore, common designs rely on front-end wireless sensors and low-latency computation offloading [2].

This perspective particularly reflects navigation systems for blind and visually impaired (BVI) individuals [3]. The purpose of such systems is to promote self-sufficiency when finding a destination point in a city, campus, etc., through orientation and mobility assistance.

The most common navigation systems verbally guide the user throughout a route, relying on GNSS or a beacon infrastructure to locate the user outdoors and indoors, as seen in Lazzus, BlindSquare and NavCog smartphone apps. This is usually enriched with data of nearby points of interest recorded in remote sources, such as Google Places or OpenStreetMap. Conversely, other systems provide assistance in close-range scenarios, allowing the user to identify and react to potentially hazardous elements in their paths. The usage of sensors varies, but it usually includes cameras and depth sensors to perform simultaneous locating and mapping (SLAM) [4] or object identification, e.g., applying pre-trained convolutional neural networks such as YOLO [5,6], while benefiting from edge-computing schemata [4,6].

Even within the field of perceptual and cognitive assistance, system delay constitutes a critical technical requirement for navigation purposes. For instance, system responses in the order of tenths of a second might be needed so that the user can avoid moving elements in their path. However, there is another aspect that might impose even stricter timing constraints: the human–machine interface.

As the amount of available data grows, the need for effective and efficient media to provide spatial information to the user has become more apparent. The study of low-level, nonvisual human perceptual and cognitive processes has revealed another foundational element of navigation systems. In this context, the pioneering work of Bach-y-Rita et al. introduced a new approach: the images captured by a camera could be encoded as tactile stimuli in a test subject’s skin, allowing for the recognition of remote elements. This assimilation of information related to a specific sensor modality through another was coined as “sensory substitution,” although later studies argued against the suitability of the term [7].

For that distal attribution to happen, the users needed to manage the camera themselves to identify the “contingencies between motor activity and the resulting changes in tactile stimulation” [8]. In line with this, recent theories view human cognition as deeply grounded in action [9]. Particularly, the sensorimotor contingencies (SMCs) theory revised sensorimotor coupling as a bidirectional process—in opposition to classic unidirectional models—that is constitutive of perception.

Overall, in navigation assistance for BVI individuals, new spaces of human–world interaction are created that ultimately rely on sensorimotor coupling. Due to their technical implementation, the navigation systems are subjected to multiple sources of delay, e.g., propagation speed, spectrum saturation, traffic congestion, computing time, etc., most of which are aggregated by the end-to-end communication quality of service (QoS).

In line with this, the degradation of the QoS, emphasizing communication latency and jitter, is expected to have a negative impact on such human–machine interfaces, and in turn system performance.

This design approach could be extended to future perceptual and cognitive assistance design. In that regard, several methodologies grouped under the term “quality of experience (QoE) [10] have been developed to assess proper operation of the system from the perspective of the user.

For instance, the ITU’s recommendations when reproducing 360° video in head-mounted displays [11] include asking the test subject about sickness symptoms or even evaluating exploratory movements with eye tracking. Additionally, performance-based tests are suggested for task-oriented applications, e.g., recognition of individuals from a video stream for surveillance purposes [12]. In general, subjective QoE analysis allows the development of system-performance prediction models from user behavior and objective QoS measurements (e.g., [13,14]). However, no prior work was found in relation to immersive human–machine interfaces or navigation assistance as mechanisms to counter visual disabilities.

In this context, the present contribution focuses on analyzing potential constraints of system delay in navigation assistance that derive from the human–machine interface.

To that end, a mixed-reality tool was developed to perform simple perceptual and cognitive experiments with sensory substitution devices (SSDs) under different system-delay scenarios. The experiments were designed taking into consideration the main purpose of a navigation system, i.e., assisting in orientation and mobility tasks throughout a route. In line with this, the measured variables are related to key user navigation performance indicators.

## 2. Materials and Methods

To analyze the impact of system delay on navigation systems, two SSDs were tested in immersive virtual environments. The system delay was degraded according to previously modeled stochastic processes. Finally, different user-performance measurements were recorded.

In the following subsections, updated versions of the navigation systems’ test-bench tools Virtually Enhanced Senses (VES) [15] and VES-PVAS SSD [16] are briefly presented. Thereafter, the three perceptual and cognitive tests conducted are described.

### 2.1. The Virtually Enhanced Senses System

The VES system is a wireless, mixed-reality platform developed to implement and test complete navigation systems (Figure 1). It immerses the user in virtual or previously scanned real environments through a visual–inertial motion-capture system. Additionally, VES allows the implementation of a wide range of sensory substitution devices, degrading network quality of service (QoS) according to previously modeled stochastic processes, and taking user behavioral data regarding navigation performance.

The controlled QoS degradation is a new feature embedded in a custom communication stack over UDP/IP. Following a publisher-subscriber pattern, streaming data such as user motion or motor driving signals is shared among devices as time-stamped packages. Each device queues the incoming stream data and discards or delays each package according to a previous simulation of end-to-end communication. Therefore, it supports simultaneous data streams with different conditions of jitter and packet loss, which in turn results from the simulation of several protocol stacks under specific conditions of traffic load, spectrum occupation, etc.

Nevertheless, this solution has two drawbacks. Firstly, it operates over existing wireless communications, and therefore there is a QoS baseline. Secondly, the implementation of VES over a non-real-time OS such as Windows might include unavoidable jitter. However, this approach is considered enough for testing purposes if the time resolution is maintained over 20–30 ms.

### 2.2. VES-PVAS Sensory Substitution

VES-PVAS is a virtual SSD based on the Virtual Acoustic Space (VAS) project. Following its predecessor, this SSD captures the 3D surfaces within a virtual camera’s field of view (FoV) and sequentially reproduces short-duration spatial virtual sound sources over it, i.e., stereo pixels (Figure 2). These stereo pixels take advantage of natural hearing perceptions of distance or material composition or even the size of an element through the number of sound sources triggered.

This kind of SSD design approach is advantageous in terms of both processing load and benchmarking, as all user–environment interactions are fully characterized by a limited set of moving spatial points that trigger stimuli—e.g., stereo pixels—and the user’s motion. In cases of mixed or virtual reality approaches, the graphical rendering is not strictly required. The density of points can be adjusted dynamically according to the required spatial resolution.

The previous VES-PVAS allowed configuration of the FoV, the N × M matrix of spawning points for the stereo pixels, the detection range, and the seed for the pseudo-aleatory spawn sequence criteria of the stereo pixels within the FoV. In addition to this, the current version includes the following.

Configurable segmentation. All elements in the virtual environment can be freely grouped and mapped to specific configuration and spawn rules for the stereo pixels. This feature builds on previous attempts to embed Gestalt-like laws of visual perception in SSDs [8,17].Figure–background discrimination. Building on the configurable segmentation feature, the elements within the FoV are identified as “focused” or “not focused.” Specifically, only the element that occupies the center of the FoV (Figure 2) is considered “focused.” Thereafter, all the stereo pixels triggered by focused and unfocused elements are added a constant volume gain accordingly.Peripheral and foveal vision. Two overlaying VES-PVAS SSDs were configured: the first one uses a relatively wide FoV and low spatial resolution, whereas the second one is set with a narrow FoV and high spatial resolution, as shown in Figure 2.

In early tests, the elevation of the stereo pixels seemed difficult to locate through the head-related transfer function (HRTF) module. Therefore, taking advantage of an altitude-to-pitch cross-modal correspondence, the stereo pixels’ sound pitch was modulated according to its relative altitude within the FoV. This is modeled by the following expression:$$A_{out}\left(f\right)=A_{in}(f*[1+\frac{1}{2}\tan\alpha ])$$
where *A*(*f*) is the audio signal spectrum and α the relative altitude within the FoV in degrees.

### 2.3. Experimental Procedures

As described in previous sections, the objective is to assess the impact of system delay in navigation systems for BVI individuals. Overall, the aim is to obtain objective measurements related to navigation performance and analyze how these measurements vary under different scenarios of system delay. In this regard, the first challenge consists in choosing appropriate user-testing methods [18].

Starting from the premise of active perception, test subjects must be allowed some degree of freedom when interacting with the environment. This would be key for the user to develop sensorimotor strategies [19], which are extended to locomotion if the spatial features of the environment cannot be apprehended from a single point. However, complete navigation experiments that require moving through an unknown environment with an artificial SMC apparatus, in this case the SSD, are sensitive to multiple variables that would outweigh the role of the system delay.

Given this context, the methods described herein were conceived as unitary tests. Simple perceptual and cognitive processes useful for navigation are evaluated in virtual environments with only 2–3 elements that can be perceived from a single vantage point. Three tests were conducted: relative width estimation, symbol discrimination, and finally, search and focus a single moving element.

The focus of the present contribution is placed on the temporal restrictions on system delay deriving from the sensorimotor coupling. To that end, an artificial delay was introduced in all three tests between the user’s motion and the resulting haptic/acoustic stimuli. Early tests revealed that approximately 500 ms produced noticeable effects in the measured performance indicators. In line with this, three models of system delay were tested: no delay, constant delay of 500 ms, and a variable delay following a Gaussian distribution of N (500, 100) ms.

No prior SSD training was done. Instead, the experiments were designed with increasing difficulty (Table 1), taking advantage of the relatively low cognitive load and intuitiveness of the SSD under test. This approach has a double purpose: firstly, it serves to homogenize the initial conditions for all users, and secondly it allows observation of the learning curve and any hypothetical adaptation to sensorimotor coupling impairment as the system delay degraded.

The tests were conducted on a heterogeneous sample of 22 normal-sighted and 3 BVI users aged 19–72 years, with a male-female ratio of 17-8. In the BVI group, the first test subject lacked peripheral vision, while the remaining subjects were completely blind. Although the sample size was not sufficient to get any statistical significance, it served as a first step in the study of the users’ behavioral patterns.

The instructions for the tests were provided to the normal-sighted users in a video tutorial. BVI users were given a verbal explanation.

Further details on the three tests are included in the following subsections.

#### 2.3.1. Relative Width Measurement

This test addresses the field of relative distance estimation. It starts from the premise that this is a basic element in the development of mental representations of space, and it holds a key role in both navigation and mobility tasks. Nevertheless, recent research questions the extent of spatial knowledge that is required for navigation [20].

The SSD used, from now on referred as “virtual cane,” consists of a virtual proximity sensor with a relatively narrow FoV (~10°) that triggers haptic stimuli in the user’s hand once an element enters its detection range (Figure 3).

This type of SSD has been used extensively since the first developed electronic travel aids, including a variety of commercial products. This extends to currently available devices and ongoing research on BVI navigation. Taking advantage of its relatively low cognitive load, the information provided is usually enriched with auditory feedback [21] or by relying on multiple sensors distributed over the user’s body [22].

Although the virtual cane encodes distance measurements as vibration intensity, in the following experiments, the distance variations are considered negligible; therefore, the SSD will only activate or deactivate the haptic actuator (ERM motor) according to the presence or absence of elements at the pointed direction. Additionally, in the current configuration of the motor driver, the start and brake times are measured under 100 ms, adding to the system delay.

As for the methods to evaluate distance estimates, a simple paired comparison was preferred, as no external metric was required. The distance measurements would need few sequences of movements, as well as low usage of working memory and inferential processing. Thus, the system delay was expected to play a major role in the obtained data, which were reduced to hit rates that could be analyzed from an above-chance perspective.

Once the test begins, the user is immersed in a virtual scenario with two rectangles 5 m in front (Figure 3). As for their dimensions, three variations were presented to study a possible trade-off between sensorimotor delay and spatial resolution when developing a mental representation of the environment. With a base dimension of 0.7 × 3.5 m, three relative widths were tested: 1.3, 1.45 and 1.7, which corresponds to 0.9, 1 and 1.3 m respectively–. For each comparison, the rectangles swapped positions at random. Finally, the time to answer was also recorded to observe possible variations under different test conditions.

#### 2.3.2. Symbol Discrimination

The user is asked to use the virtual cane to identify a symbol from a known set (Table 2), provided as 3D-printed embossed images. The purpose is to check whether specific spatial features from the environment can be recognized, e.g., building on distance estimations made with an SSD. The virtual scenario is the same as in the previous test (Figure 3).

Originally, the symbols were designed to check if the user could discriminate the number of elements and their relative dimensions with simple horizontal and vertical movements. This was useful to evaluate potential “proximity” between symbols from the perspective of the user estimations. This in turn relates to the specific features of the sensorimotor apparatus (SSD).

Early tests were conducted to adjust the size and content of the set. Finally, the six symbols included in Table 2 were considered enough for this contribution. The symbol under **E** was included to increase the difficulty, as it forces higher-precision movements for effective discrimination.

Once the test starts, the user is presented with a random permutation of all 6 elements of the set. This is repeated for all sensorimotor delay scenarios. Again, the experiments follow a common order for all users (Table 1), in which the system delay is degraded progressively. The hit rate and time to answer are recorded for all scenarios.

#### 2.3.3. Search and Focus on a Moving Element

Finally, the third test was conceived as a performance analysis of SSD that encodes stream data from a camera. This constitutes one of the main families of SSD, which includes well-known visual-to-auditory SSD projects such as vOICe, EyeMusic [23], or VAS, as well as visual-to-tactile SSD, e.g., BrainPort [24], etc.

It is noted that these systems are especially prone to incur an additional system delay due to bandwidth differences between sensory modalities. This point is exemplified by vOICe and EyeMusic, in which low-resolution images are converted into audio signals with durations in the order of seconds. This acts as a baseline sensorimotor coupling impairment, which can be reduced at the cost of spatial resolution.

Analogously to the visual sense, this family of camera-based SSD provides simultaneous perception of different elements within the FoV. In navigation tasks, one of the main advantages is that the user is informed about the presence and relative position of elements in a relatively large area. In line with this, navigation performance is evaluated with a simple perceptual test: pointing to a single element.

The user is asked to keep track of a moving element that follows a predefined sequence of positions unknown to the user. For this test, the equipment consists of a VR headset with a 6 DoF MoCap module and a couple of headphones. The specific configuration of VES-PVAS is included in the following table (Table 3).

The sequence is composed of 9 positions arranged in a 3 × 3 matrix, as shown in Figure 4. The element covers all 8 possible trajectories, starting from the center. Once the test starts, the element stay in place until the user maintains their focus steadily for 3 s through VES-PVAS. Thereafter, it moves to the next position within the gray surface, i.e., screen, and so on. This cycle is repeated for all 3 scenarios of sensorimotor delay. Finally, the time to focus (TTF) and the user’s gaze trajectories are recorded as objective performance variables.

The positions of the element were chosen according to the following criteria.

The element should always be perceivable by the user, i.e., stay within the FoV. This eliminates the need for exploratory movements, which would otherwise constitute a parasite effect included in the measured variables.The test must favor intentionality in the user–environment interaction. In line with this, it was concluded that the element motion should include uncertainty in at least two axes. This serves to avoid focusing the element by chance or with simple sweeping movements independently of the SSD feedback.The SSD feedback is different when providing information regarding the X and Y relative position. Therefore, various combinations of X–Y trajectories should be included to assess any potential impact on the performance indicators.

The users are expected to unfold the SSD feedback without previous training, as experienced in previous work [11]. Thereafter, the TTF is directly related to performance: it would increase in scenarios with higher difficulty, e.g., under system delay degradation. Conversely, it would decrease with user experience.

On the other hand, the gaze tracking shows the user motion when focusing on an element. It is considered that in the best case, it would follow a straight line from the initial to the ending position. Under system delay, a deviation is expected.

## 3. Results

In total, 25/25, 23/25, and 23/25 of the users were able to complete the first, second, and third tests, respectively. This accounts for the intuitiveness of the SSD implemented, as well as the embedding of SSD training in the user testing. On the other hand, the results of the BVI and normal-sighted groups showed no significant difference. Therefore, the following data aggregate all test subjects indistinctly.

The hit rate and time to answer for the relative width measurement and symbol discrimination are shown in Figure 5. This figure includes the average results from the first two tests, divided into experiments 1–12, as described in Table 1.

For all 12 experiments making up the three tests (Table 1), the results were above chance. In the first test (1–9), the hit rate in all three paired comparisons denoted a general trade-off between spatial resolution and system delay. Conversely, the time to answer showed a significant increment only in symbol discrimination as the delay degraded.

All users reported that the elements in the virtual scenario “moved“ as the system delay increased, and some even related that to their own movement, but none seemed to associate it with a delay in the stimuli triggering.

In relation to the second test, the questionnaire answers are gathered in the following confusion matrices (Figure 6). On top of the hit rate, these matrices show the most common symbol-guess errors made by the users. From left to right, each of the matrices corresponds to a different system delay scenario.

As could be anticipated, each symbol showed different hit rates and statistical proximity to the other symbols in terms of user estimations. However, despite showing analogous geometric features, the pairs A–B (mirroring) and C–F (rotation) results differed. On the other hand, the element E exhibited the worst performance results for all three delay scenarios.

Outside the methods specified, sensorimotor strategy development was also observed. When using the virtual cane, almost all users seemed to measure distances through vibration duration as they moved the cane at a constant speed along the X or Y axis. The accuracy of those movements showed key importance throughout the tests. It could even be used to advance the user answers to the questionnaires.

Figure 7 shows the average gaze-tracking heatmap of the user in test 3. This heatmap is a 2D histogram of the users’ gaze as it moved within the gray screen shown in Figure 4. This figure included the starting and ending position of the moving element as white and red squares, respectively. Finally, the 24 graphs correspond to all 8 × 3 combinations of element trajectory and system delay.

In all experiments, the “gaze” was always centered close to the target. However, the gaze was relatively dispersed in diagonal trajectories and degraded sensorimotor delay scenarios.

After a few experiments, the users tended to wait for the peripheral stereo pixels to trigger. In line with this, the heatmap shows higher intensity at the center. This result shows an adaptation to the relatively low temporal resolution of camera-based SSD. In particular, the current configuration of VES-PVAS requires approximately 2.4 s to cover all spawn points of the stereo pixels. Nevertheless, this SSD offers a time-resolution trade-off by distributing the positions of the stereo pixels’ spawning sequence, based on the traditional interlaced video. This feature serves to accelerate the feedback of relatively large elements within the FoV.

These data are complemented by the TTF, that is, the time required by the test subjects to focus on the moving element. In turn, this time has been divided according to two different events: “first focus” and “maintained focus,” i.e., that which triggers the element motion. This can be observed in the following figure.

In Figure 8, the exploratory head movements can be noted: the space between events shows “spikes” in VES-PVAS rotational speed as the user tries to center the element in the foveal region. Once the user focuses on an element, the rotational speed diminishes as he/she tries to maintain it at the center of the FoV.

The next figure presents the average (Figure 9, left) and standard deviation (Figure 9, right) values of the TTF corresponding to all eight trajectories of the moving element, following the order of the user experiments (Table 1).

As can be observed, with no prior explanation regarding VES operation, the first position was difficult to detect: it exhibits large average and dispersion values. After that, the TTF converged for all trajectories and users, given the relatively low standard deviation. Once the half-second delay was added, the TTF increased abruptly and with high variance among users (Figure 9, right). Nevertheless, it decreased over time, and stabilized even after the addition of jitter.

Finally, the number of “element-focused” events until the element moved to the next position, i.e., focus attempts (Figure 8), is included in Figure 10. In contrast to the TTF, it increased constantly as the system delay degraded, with no evident signals of user compensation.

## 4. Discussion

Overall, the under-second system delay added in the sensorimotor coupling was enough to noticeably degrade the modeled navigation performance indicators. This applies to both the SSD and the three conducted tests. Nevertheless, some sort of compensation mechanism was recognized in VES-PVAS.

The first remarkable point is the trade-off between system delay and spatial resolution observed from the hit rates in the first test. The well-known virtual cane approach denoted poor performance in terms of relative distance estimation and identification of primitive figures, which was more apparent as the delay increased and the spatial resolution lowered. Specifically, the latter effect could be observed from the users’ poor discrimination rates of figures of relatively little dimension differences. However, it should be noted that the actuator that triggered the tactile stimuli was an ERM motor driven by pulse width modulation PWM signals. Therefore, the ~100 ms maximum time for the start and brake times are noticeable by the user, and act as a low-pass temporal and spatial filter when the user interacts with the environment.

Additionally, the users reported that the element moved, translating the internal effect of an impaired sensorimotor coupled with a distorted representation of the environment. Some of them even accelerated their movements to catch the presumably moving element. This suggests that the temporal requirements for navigation purposes, and specifically for relative position estimation, are even more restrictive.

On the other hand, the experiments endorse the hypothesis that the acquisition of spatial information is exteriorized through the user’s interaction with the environment. As described in the previous section, the questionnaire answers could be anticipated after careful examination of the movements of the user within the virtual scenario and the corresponding stimuli generation. This approach to the analysis of sensorimotor strategies could be used to further improve SSD design, propose objective performance parameters, or even adjust the human-machine interface with human-in-the-loop strategies.

As for the third test, one of the most remarkable results is the fast-learning curve in the usage of VES-PVAS. Almost all users were able to focus the elements with no prior training or explanation regarding the sound patterns. Furthermore, the gaze was concentrated in the surroundings of the target for all combinations of element trajectories and sensorimotor delay scenarios.

After adding sensorimotor delay, a second learning curve was seen in which the users adapted their movements according to the new perceptual conditions. This seems to occur in opposition to the results of the virtual cane SSD, in which the internal sensorimotor impairment was translated to external sources and no compensatory behaviors could be observed from the data gathered.

## 5. Conclusions

The present contribution points out novel timing restrictions of system delay when providing navigation assistance to BVI individuals. These restrictions are derived from human perceptual and cognitive processes, and in turn sensorimotor coupling. Consequently, this is another factor to be taken into consideration in future networked designs in which end-to-end communication delay plays a major and unavoidable role.

The results suggest that there are not hard timing restrictions, but a trade-off between the detail and amount of spatial information that can be provided and the level of degradation of the system delay. This trade-off could even benefit from training, as the test subjects showed a learning curve even under a half-second system delay.

Finally, this contribution proposes a novel approach to human–machine interface design based on objective user-performance parameters gathered from an artificial SMC apparatus. Specifically, it was used to assess the impact of motion-to-photon latency in nonvisual, immersive human–machine interfaces. Overall, this is expected to be useful in the development of future devices for perceptual and cognitive assistance.

## Figures and Tables

**Figure 1 sensors-23-03219-f001:**
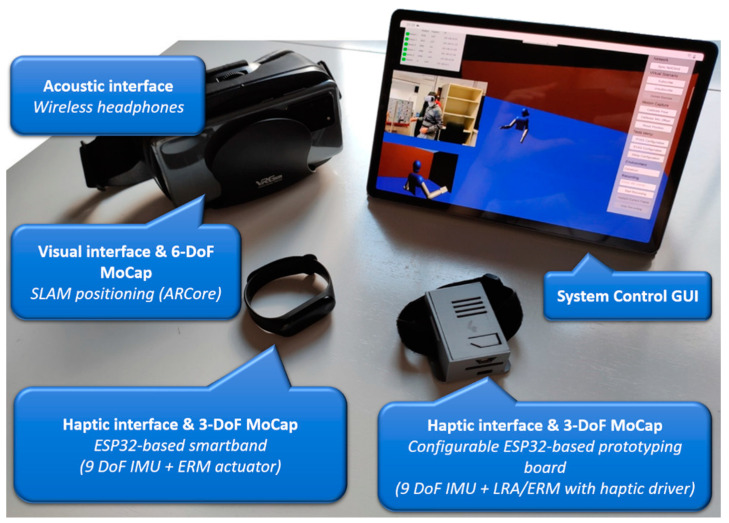
The VES system (the image in the tablet is a capture of the GUI and a photo of the user).

**Figure 2 sensors-23-03219-f002:**
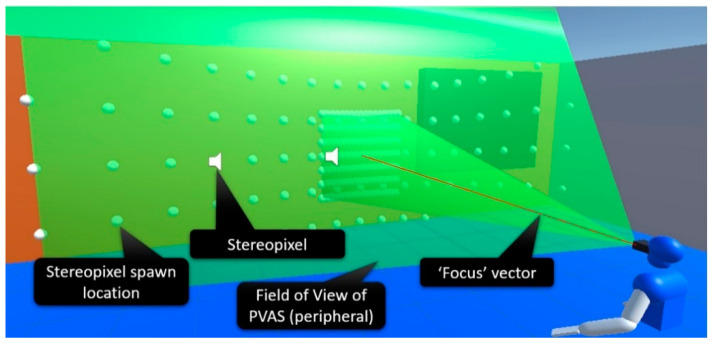
VES-PVAS sensory substitution system.

**Figure 3 sensors-23-03219-f003:**
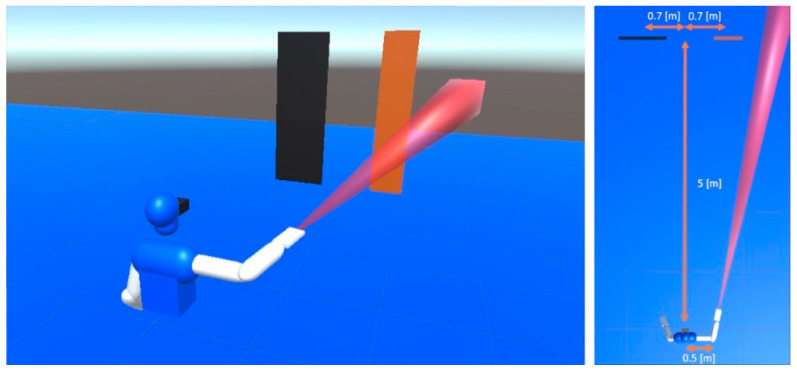
Test of relative width with a haptic SSD. The detection area of the SSD is in red.

**Figure 4 sensors-23-03219-f004:**
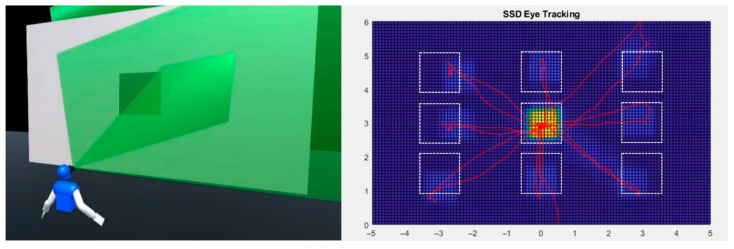
Test of search and focus a moving element. In the right image, mock SSD eye tracking is presented.

**Figure 5 sensors-23-03219-f005:**
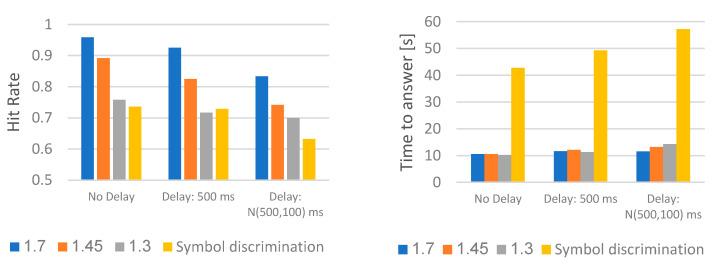
Average hit rate and time to answer in the relative width measurement and symbol identification tests.

**Figure 6 sensors-23-03219-f006:**
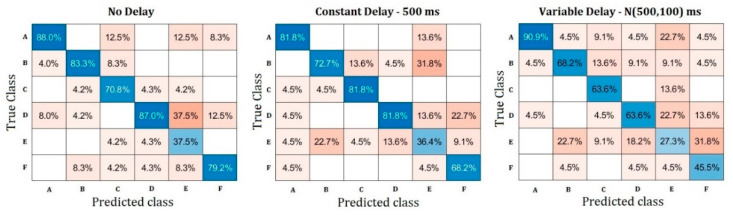
Confusion matrices in the symbol identification test.

**Figure 7 sensors-23-03219-f007:**
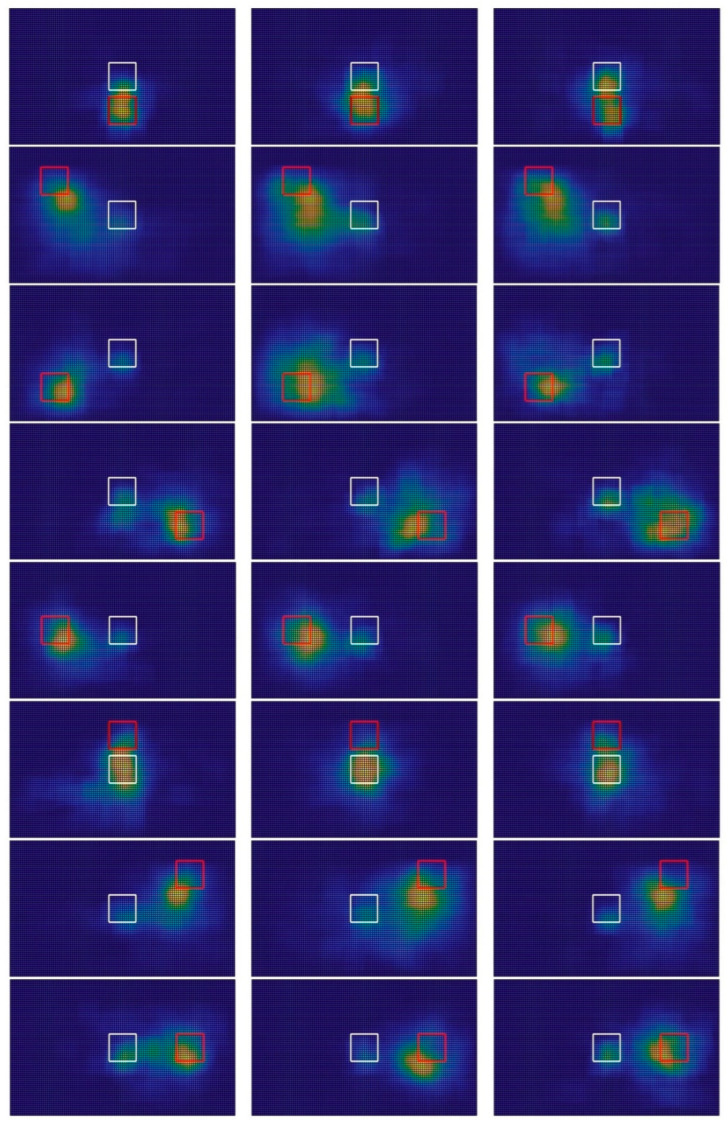
VES-PVAS average foveal gaze-tracking heatmaps of test 3. Each column corresponds to all element trajectories, starting from the center, under the conditions of no delay, 500 ms delay and N (500, 100) ms delay. The white and red boxes represent the starting and ending position of the element.

**Figure 8 sensors-23-03219-f008:**
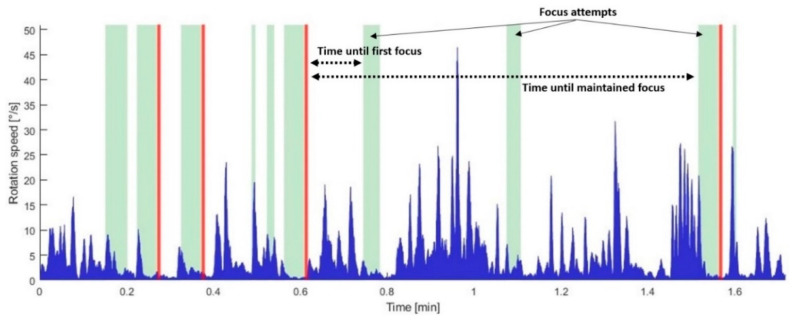
Gaze motion data of a single user in the third test. It includes rotation speed of VES-PVAS FoV (blue): “element focused” (green bars) and “element moved” events (red bars).

**Figure 9 sensors-23-03219-f009:**
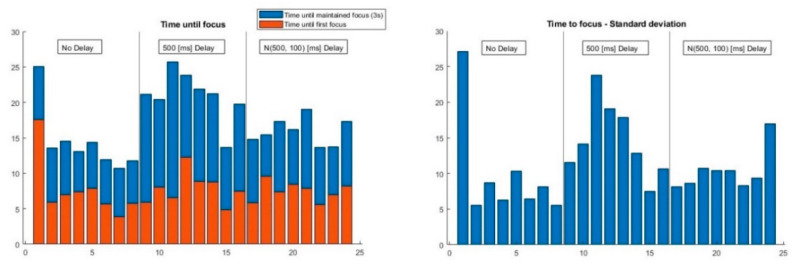
Time inverted until the element was steadily focused (3 s) per trajectory: average (**left**) and standard deviation (**right**) values.

**Figure 10 sensors-23-03219-f010:**
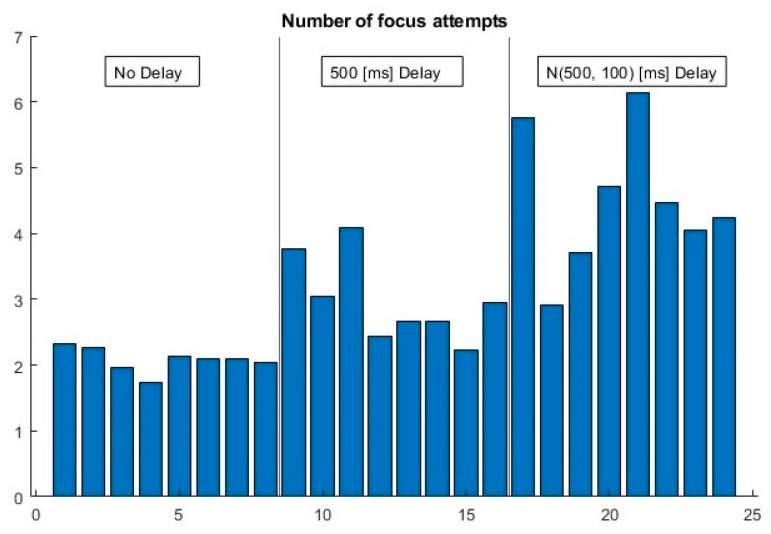
Number of times that the element was focused until it moved to the next position.

**Table 1 sensors-23-03219-t001:** Order of the experiments.

Order of The Tests		Sensorimotor Delay		
		None	500 ms	N (500, 100) ms
Relative width estimation	Ratio 1.70	1	4	7
	Ratio 1.45	2	5	8
	Ratio 1.30	3	6	9
Symbol discrimination		10	11	12
Search and focus on moving element		13	14	15

**Table 2 sensors-23-03219-t002:** Set of symbols presented to the user.

Symbol	A	B	C	D	E	F
Figure				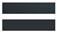	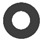	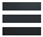
Dimensions [m]	4.5 × 2.5	4.5 × 2.5	4.5 × 2.5	4.5 × 2.5	4 × 4	4.5 × 2.5

**Table 3 sensors-23-03219-t003:** VES-PVAS configuration of the third test.

PVAS Configuration	Peripheral	Foveal
M	17	17
N	7	7
FoV_x_	100°	20°
FoV_y_	50°	20°
Detection distance	20 m	20 m
Seed	43	43
Period	20 ms	20 ms
“Focused” volume gain	−1 dB	0 dB
“Not focused” volume gain	−3.5 dB	−9 dB

## Data Availability

The raw results of the user tests are available online at (DOI: 10.17632/zzswztx4cz.1) and http://elb105.com/ves/ (accessed on 14 March 2023).

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
