# Peer review of "Network QoS Impact on Spatial Perception through Sensory Substitution in Navigation Systems for Blind and Visually Impaired People"

_sensors, 2023, doi:10.3390/s23063219_

Round 1

Reviewer 1 Report

The authors proposed the article titled Network QoS impact on Spatial Perception through Sensory Substitution in Navigation Systems for Blind and Visually Impaired People”.  The following comments should be incorporated into the manuscript:

1.      Figure 1 is not clear therefore, it should be improved as per guidelines of MDPI.

2.      Results and discussions are not clear. It should be explain in clear.

3.      Novelty is not clear in the manuscript.  

4.      The comparative analysis should be done with existing and novel work.

5.      Conclusions should be improved as per the results obtained.

6.      All figure quality must be improved.

Author Response

(The response is attached as a pdf file)

Reviewer 2 Report

The paper searches for temporal constraints in navigation systems for blind and visually impaired people due to human-machine interfaces.  The area of research is object of great interest in recent years due to the increasing number of smart devices used in the every day life and their importance for the telemedicine/telecare/telehealth areas. Network QoS is important parameter of the service delivered.

The Introduction is extensive and well-written and introduces the reader to the basic concepts in the paper, such as: navigations systems for blind and visually impaired, various smartphone apps, simultaneous locating and mapping, sensorimotor contingencies theory, aggregated by the end-to-end communication quality of service (QoS).  All references are up-to-date and related to the  corresponding concepts. The only exception is the end-to-end QoS   in telecommunication systems/networks. The authors should include references to the documents of the International Telecommunication Union regarding the QoS concept. For example,the QoS regulations (ITU-T Supp. 9 of E.800 Series), QoE requirements for real-time multimedia services over 5G networks (https://www.itu.int/pub/T-TUT-QOS-2022-1); ITU-T Recommendation ITU-T P.10/G.100  (11/2017). Vocabulary for performance, quality of service and quality of experience, etc.

 Also,  the authors should address the dependence between QoS and Quality of Experience (QoE) in overall telecommunication networks. In recent years there are important studies on QoS and QoE. For example, the importance of the normalization of the concepts, normalization of the indicators’scales, etc. are discussed in the paper:

S. A. Poryazov et al., "Overall Model Normalization towards Adequate Prediction and Presentation of QoE in Overall Telecommunication Systems," 2019 14th International Conference on Advanced Technologies, Systems and Services in Telecommunications (TELSIKS), Nis, Serbia, 2019, pp. 360-363.

Also, there is a paper of Reichl from 2010:

P. Reichl, S. Egger, R. Schatz and A. D'Alconzo, "The Logarithmic Nature of QoE and the Role of the Weber-Fechner Law in QoE Assessment," 2010 IEEE International Conference on Communications, Cape Town, South Africa, 2010, pp. 1-5.,

 which exaplains the logarithmic nature of the QoE and the Weber-Fechner Law. Such more recent studies on the relation between QoS and QoE should be included in the Introduction/the literature review.

Section 2, discusses the Materials and Methods used in the study. It is very well structured and clearly presents the materials and methods. It is divided into three subsections: The Virtually Enhanced Senses System, VES-PVAS Sensory Substitution and Experimental Procedure.

The VES system is illustrated on Figure 1. The reference to that figure  on line 100, must have the first letter capitalized.  The term “quality of service” on line 102, and elsewhere in the paper, should be replaced by the already introduced abbreviation QoS.  The picture quality must be improved as the text inside the picture is not readable.

The VES-PVAS Sensory Substitution System is illustrated on Figure 2. Its elements are explained well.  The last sentence in subsection 2.2 must be edited in the following way: “This is modelled by the following expression:” followed by the equation  on line 157, which must be numbered and followed by “where ‘A(f)’ is the audio signal spectrum and ‘α’ is the relative altitude within the FoV in degrees”.

The experimental results are presented in subsection 2.3. The results are clearly presented and the approach is scientifically sound.  The tests were conducted to 22 normal-sighted and 3 BVI users aged between 19-72 years old, with a male-female ratio of 17-8. The authors should address the following questions:

11)     Isn’t the sample size too small to draw a  conclusion?

22)     What would be the effect of the 17-8  ratio of mles-females?

The test of the relative width with a haptic SSD is illustrated on Figure 3. The reference to that figure on line 205 should become “Fig. 3”.

The numerical results presented in section 3 seem to be correct and are very well presented.

The quality of Figures 6,8, 9, 10 must be imporved as the text in the pictures is not readable.

The discussion of the results is sufficient and the conclusions are supported by the results.

Overall, the paper is interesting and represent a valuable study. I recommend that the paper be published once the authors address adequately the above remarks.

Author Response

(The authors gave the same response as above.)
